# Recent Advances in the Use of Molecular Methods for the Diagnosis of Bacterial Infections

**DOI:** 10.3390/pathogens11060663

**Published:** 2022-06-08

**Authors:** Elisabetta Gerace, Giuseppe Mancuso, Angelina Midiri, Stefano Poidomani, Sebastiana Zummo, Carmelo Biondo

**Affiliations:** 1ASP (Azienda Sanitaria Provinciale), 90141 Palermo, Italy; elisag74@live.it; 2Department of Human Pathology, University of Messina, 98125 Messina, Italy; mancusog@unime.it (G.M.); amidiri@unime.it (A.M.); stefano.poidomani@gmail.com (S.P.); zummo@unime.it (S.Z.)

**Keywords:** multidrug-resistant bacteria, antimicrobial resistance, ESKAPE, genotyping methods

## Abstract

Infections caused by bacteria have a major impact on public health-related morbidity and mortality. Despite major advances in the prevention and treatment of bacterial infections, the latter continue to represent a significant economic and social burden worldwide. The WHO compiled a list of six highly virulent multidrug-resistant bacteria named ESKAPE (*Enterococcus faecium*, *Staphylococcus aureus*, *Klebsiella pneumoniae*, *Acinetobacter baumannii*, *Pseudomonas aeruginosa*, and *Enterobacter* species) responsible for life-threatening diseases. Taken together with *Clostridioides difficile*, *Escherichia coli*, *Campylobacter* spp., (*C. jejuni* and *C. coli*), *Legionella* spp., *Salmonella* spp., and *Neisseria gonorrhoeae*, all of these microorganisms are the leading causes of nosocomial infections. The rapid and accurate detection of these pathogens is not only important for the early initiation of appropriate antibiotic therapy, but also for resolving outbreaks and minimizing subsequent antimicrobial resistance. The need for ever-improving molecular diagnostic techniques is also of fundamental importance for improving epidemiological surveillance of bacterial infections. In this review, we aim to discuss the recent advances on the use of molecular techniques based on genomic and proteomic approaches for the diagnosis of bacterial infections. The advantages and limitations of each of the techniques considered are also discussed.

## 1. Introduction

The rapid global spread of multidrug-resistant bacteria poses a serious threat to public health worldwide [1]. Critical-priority bacteria, as defined by the World Health Organization (WHO), include a group of life-threatening nosocomial pathogens known as “ESKAPE”, an acronym indicating the names of these bacteria and their ability to evade the antimicrobial activity of antibiotics [2]. The group of multidrug-resistant ESKAPE bacteria “for which new antibiotics are urgently needed”, as highlighted by the WHO, are: *Enterococcus faecium*, *Staphylococcus aureus*, *Klebsiella pneumoniae*, *Acinetobacter baumannii*, *Pseudomonas aeruginosa*, and *Enterobacter* spp. [3]. These pathogens, together with *Clostridioides difficile*, *Escherichia coli*, *Campylobacter* spp., (*C. jejuni* and *C. coli*), *Legionella* spp., and *Salmonella* spp., are the most common causes of nosocomial infections [4,5]. These pathogens exhibit drug resistance through several mechanisms, including active site modification, drug inactivation, efflux pump overexpression, and biofilm formation (Figure 1) [6,7,8]. As a result, these microorganisms are able to persist for long periods of time in hospital environments, becoming resistant to biocides or otherwise limiting their effects, spreading from person to person, and causing serious hospital infections [4,9,10]. The ability to produce enzymes such as ESBLs and carbapenemases, which can inactivate antibiotics of last resort, have contributed greatly to the rapid spread of the Gram-negative members of the *ESKAPE* group, especially in intensive care units (ICUs) [11,12]. For the same reason, methicillin-resistant *Staphylococcus aureus* (MRSA) and vancomycin-resistant *Enterococcus* (VRE) are the two members of the Gram-positive *ESKAPE* group that have been identified as two of the most important threats to patients in health-care settings [13,14]. Therefore, sensitive and specific microbe detection tests are needed to enable the rapid implementation of infection control practices. In recent decades, molecular diagnostics have developed considerably with the development of hybridization techniques that, through short sequences of nucleotide bases labeled with fluorescent tags (known as probes), are able to detect the presence of particular DNA sequences of interest (Appendix A). Numerous commercial kits are available for the rapid detection of different pathogens and their associated AMR genes directly from clinical specimens (Figure 2). However, as discussed below, all these molecular diagnostic techniques have both advantages and limitations compared to classical methods. According to the 2019 Centers for Disease Control and Prevention (CDC) report, resistance to essential antibiotics was increasing in seven of the eighteen germs originally reported in 2013 list [15]. The new report includes a new urgent threat posed by carbapenem-resistant *Acinetobacter* spp. causing pneumonia, bloodstream, wound, and urinary tract infections, primarily in hospitalized patients [15,16]. As mentioned above, *Staphylococcus aureus* is commonly found in the skin, but it is also a frequent cause of infection in catheterized patients [17]. *Pseudomonas aeruginosa* causes infections especially in hospital patients with weakened immune systems and can lead to worsening of previous lung diseases [18]. *Enterobacter* spp. can cause serious nosocomial infections, including blood and urinary tract infections, that are resistant to all antibiotics except for tigecycline and colistin [15,19]. These are also the last options against carbapenem-resistant hypervirulent *Klebsiella pneumoniae* [15,20]. Enterococci spp are major causes of hospital-acquired infections [21]. If these bacteria spread from the intestines, where they normally live harmlessly, to other parts of body, they can cause more serious infections, including bloodstream, surgical site, and urinary tract infections [22]. According to the 2019 CDC report, *E. faecium* caused 54,500 hospitalizations and 5400 deaths in 2017 in the US [23]. A broad range of enteric pathogens, including *C. difficile*, *Campylobacter*, and *Salmonella* species, are the most common causes of antibiotic-associated infectious diarrhea [24]. *C. difficile* is the pathogen responsible for the largest number of healthcare-associated infectious diarrheas and pseudomembranous colitis worldwide [24,25]. Cases of nosocomial legionellosis have increased substantially in recent years, with a lethality rate for healthcare cases exceeding 50% [26]. In addition, the risk of disease increases dramatically if the germ is found in certain areas within the hospital that are designed to closely monitor and treat patients with life-threatening conditions, such as intensive care units [27].

## 2. Detection and Identification of Bacterial Pathogens

The difficulty in identifying bacterial pathogens and their associated AMR genes, due to the lack of rapid diagnostic methods, has long been a problem that, in the clinical setting, has often been overcome using broad-spectrum antibiotics [28]. Furthermore, the identification of bacterial pathogens for early aggressive antibiotic treatment is critical to prevent chronic infections that could lead to consequences such as wound infections, pneumonia, and catheter-related bloodstream infections that can lead to sepsis [1,29]. Therefore, rapid and sensitive molecular methods used for the detection of highly virulent drug-resistant bacteria in clinical samples are needed to guide antibiotic therapy and optimize control measures. In most cases, the gold standard for microbial identification relies on phenotypic approaches, including the use of enriched and selective culture media for the isolation of pathogenic bacteria, automated biochemical testing for identification, and antimicrobial susceptibility testing to guide therapy [30].

The main disadvantages of cultivation include the selection of a microbe-specific medium, the long incubation period for the microbe to spread on the selected medium, and the time required to obtain results [31]. In contrast, molecular diagnostics can be used to detect microbes directly from clinical specimens, thereby minimizing the occupational exposure of laboratory workers to infectious agents [12,32]. Improvements in molecular techniques, along with advances in technologies associated with bioinformatics tools, have led to the development of new technologies that have revolutionized the diagnosis of bacterial infections (Figure 3). This review aims to discuss the advantages and limitations of molecular methods, based on genomic and proteomic technologies, applied to the diagnosis of bacterial infections.

## 3. Polymerase Chain Reaction (PCR) and Isothermal Amplification Methods

### 3.1. PCR

Nucleic-acid-based amplification technologies (NAATs) are high-performance tools used for the rapid and specific detection of pathogen-specific nucleic acids [33]. PCR was the first DNA amplification method, introduced by Mullis in 1985, that helps to make millions of copies of a DNA template starting from only a few molecules [34]. The PCR technique relies on thermal cycling, which consists of repeated heating and cooling cycles of the reaction to synthesize DNA. Short DNA fragments (known as primers) with sequences complementary to the target region are used to achieve exponential amplification of a target sequence, using a DNA polymerase that is stable at high temperatures (hence the name of the method) [35]. Because the identification of pathogens and their resistance mechanisms is a key challenge that must be achieved as quickly as possible to contain the further spread of antibiotic resistance, since its invention, many endpoint PCR variants have been developed to rapidly achieve these two goals [36].

### 3.2. Real-Time PCR

Due to its good sensitivity, specificity, and speed, real-time PCR, based on PCR techniques, is one of the most widely and clinically used techniques for the diagnosis of infectious diseases [36]. This technology allows real-time monitoring of the target amplification using either unspecifically intercalating fluorescent dyes or specific fluorescently labeled probes. The fluorescent signal is generated only after the probe hybridizes with its complementary target, and it is directly proportional to the number of PCR amplicons generated [28]. Although commercially distributed kits for the detection of pathogens and their resistance genes directly on biological samples are available, either false-positive or -negative results using different genes have also been previously reported [37,38,39]. This may be due to different factors, such as the rearrangement of genes within the genome and the acquisition of novel genes using horizontal transfer systems [40]. In addition, while *mec* gene identification signals methicillin-resistant *Staphylococcus aureus* (MRSA), the identification of resistance to β-lactam antibiotics in Gram-negative bacteria is more complex, because many of these bacteria contain more than one thousand β-lactamase-encoding genes [41,42]. Accordingly, different PCR platforms have focused on the most prevalent carbapenemases, such as blaKPC, blaOXA-48, blaOXA-23, blaOXA24/40, blaNDM, blaVIM, and blaIMP [43]. The main disadvantages of RT-PCR include a limited number of targets due to the limited availability of differentially fluorescent dyes. In addition, its sensitivity and specificity decrease with increasing levels of PCR multiplexing, due to the synthesis of nonspecific amplification products and the formation of primer dimers [44]. Moreover, the RT-PCR instrument requires high maintenance costs. In recent years, new developments in RT-PCR have led to the generation of a variety of isothermal amplification techniques including loop-mediated amplification (LAMP), helicase-dependent amplification (HDA), nucleic acid sequence-based amplification (NASBA), and transcription-mediated amplification (TMA) [28,35,45].

### 3.3. LAMP

LAMP is a valid and effective diagnostic test that does not require the use of expensive thermal cyclers or specialized personnel, and is cost-effective. LAMP is based on conventional PCR, but, unlike the latter, it uses a DNA polymerase with high strand displacement activity, with four primers that recognize six–eight distinct regions of the target DNA to produce a stem-loop structure of the DNA that facilitates subsequent rounds of amplification [46]. The release of pyrophosphate that follows the synthesis of the target DNA stem-loop can be detected by adding a DNA-binding dye, such as SYBR green. The ability of LAMP to generate up to 10^9^ copies of products in less than an hour makes this technique more sensitive than conventional PCR [46,47]. Commercial LAMP kits are available for the detection of resistance genes, such as: bla_KPC_ and bla_NDM-1_ in *K. pneumonia* and *A. baumannii* [47].

### 3.4. NASBA, TMA and HDA

NASBA and TMA are isothermal amplification reactions which, unlike PCR, typically use various mRNA’s as target sequences as the target sequence. These techniques are the gold standard in the diagnosis of gonorrhea and chlamydial infection [48]. HDA is another isothermal amplification method that uses the unwinding activity of a helicase to separate dsDNA into two single strands that can serve as a template for new DNA synthesis [49]. In fact, the two strands maintain a single-strand template due to the immediate binding of SSBs, which prevents the reconstitution of a double helix. HDA assays have been successfully applied to detect *Clostridium difficile*, *S. aureus*, and *N. gonorrhoeae* [45,49].

### 3.5. The BioFire FilmArray Panels

This system is a vitro diagnostic test *platform* that integrates nucleic acid extraction, reverse transcription, and nested multiplex PCR amplification, followed by a melting curve analysis [50]. This system is designed to be used with comprehensive panels that include assays for a variety of pathogens, causing respiratory viral infections, pneumonia, bloodstream, gastrointestinal infections, and meningitis–encephalitis, as well as antimicrobial resistance genes [51]. The *FilmArray* menu *comprises*: Gastrointestinal, Pneumonia, Meningitis Encephalitis, and Blood Culture Identification panels [52]. The FilmArray Gastrointestinal panel tests for 22 common pathogens, including viruses, bacteria, and protozoa, which cause gastrointestinal symptoms such as diarrheal diseases, i.e., a leading cause of child death in developing countries [53]. The FilmArray Pneumonia panel tests for 27 of the most common pathogens involved in lower respiratory tract infections and 7 genetic markers of antibiotic resistance [54]. This panel has a sensitivity and specificity for bronchoalveolar and sputum samples of >96%, enabling both the identification of the causative agents of hospital-associated respiratory infections, and also the prevention of the causative agents of secondary infections, including antibiotic-resistant strains of *S. aureus* and *K. pneumoniae* [55]. The FilmArray Blood Culture Identification panel detects 33 pathogen and 10 antimicrobial resistance genes associated with bloodstream infections [56]. For patients with sepsis, a leading cause of death in hospital patients, rapid identification of the organism from blood cultures in combination with the indication of pathogen-associated resistance genes is critical for reducing patient morbidity and mortality [57]. In addition, according to international guidelines for the management of sepsis, in order to reduce the occurrence of antimicrobial resistance, it is important to replace broad-spectrum therapy with appropriate antibiotic therapy as soon as pathogens are identified [58]. Finally, the FilmArray Meningitis Encephalitis panel is able to simultaneously test cerebral spinal fluid for the 14 most common pathogens responsible for community-acquired meningitis [59]. Meningitis affects more than one million people, and symptoms appear and progress rapidly; therefore, early diagnosis of meningitis is crucial to achieve the best treatment outcomes [59].

### 3.6. DNA Microarrays

PCR approaches are the gold standard for detecting and identifying a single gene or a limited number of genes [36]. In contrast, PCR is not a good choice for detecting a large spectrum of genes, such as those involved in antimicrobial resistance [36]. The high multiplexing ability *of* microarrays to detect and identify a vast range of genes simultaneously makes this technique noteworthy for the investigation of genetic AMR in bacteria [60]. This technology allows for the detection of genes encoding ESBLs and carbapenemases through the hybridization of oligonucleotide sequences that can select and amplify specific molecules from the sample of interest [60,61]. Some bacteria, including the ESKAPE pathogens, may have several resistance genes, which must be identified immediately to prevent the spread of multi-resistant strains. For example, ESBL genes are found on plasmids that can be easily transferred between enteric Gram-negative bacteria, such as *E. coli* and *K. pneumoniae*, which are among the most common uropathogenic bacteria that cause urinary tract infections [62,63]. A second type of multidrug resistance genes that must be recognized immediately is related to the presence of transposons carrying vancomycin resistance genes [64]. Vancomycin-resistant *S. aureus* (VRSA) is mediated by a cluster of vanA genes transferred from vancomycin-resistant enterococci [64]. These bacteria are often resistant to a wide range of antibiotics, including aminoglycosides, tetracyclines, and beta-lactam antibiotics [65]. The ability of the microarray to detect multiple multi-drug resistance genes present in a single strain can help clinicians direct antimicrobial therapy [66]. However, the microarray is not widely implemented in clinical microbiology laboratories due to the high cost of a single experiment and the complexity of the method [30,67].

### 3.7. Pulse-Field Gel Electrophoresis (PFGE), Multilocus Sequencing Typing (MLST), and Pyrosequencing 

Different methods have been described for the characterization of ESBL and carbapenemase genes, and for epidemiological investigations such as pulsed-field gel electrophoresis (PFGE), multilocus sequencing typing (MLST), whole-genome sequencing (WGS), pyrosequencing, and next-generation sequencing (NGS) [30]. PFGE is a technique used for the separation of large fragments of DNA molecules, obtained after restriction enzyme digestion, by applying an electric field to a gel matrix that periodically changes direction. Although several PFGE protocols have been developed in the past for typing different bacteria and for outbreak investigations, this technique has recently been replaced by better-performing techniques such as NGS and MALDI [68]. MLST is a strain typing system that focuses exclusively on conserved housekeeping genes to derive a combination of alleles (known as sequence type (ST)) that can discriminate isolates of bacterial species without requiring whole-genome sequencing [69]. The high cost and low discrimination power of this technique limit its use to epidemiological investigations [69]. Pyrosequencing is a bioluminescence method that measures the release of inorganic pyrophosphate. Briefly, the release of inorganic pyrophosphate (PPi) following each nucleotide incorporation event is proportionally converted, by a series of enzymatic reactions, into visible light [70]. The main drawback of this technique is that it can only sequence a short nucleotide sequence [70]. In general, the main disadvantages of these techniques are their high cost and the availability of technicians capable of executing them.

### 3.8. Whole-Genome Sequencing (WGS)

Whole-genome sequencing (WGS) refers to the determination of the complete nucleotide sequence of a genome of a microorganism in a single assay [71]. Over the last two decades, WGS has been successfully applied in different fields including: (1) the identification of the virulence factors of some pathogens; (2) the identification of the disease transmission pathway in outbreak analysis; (3) AMR profiling; and (4) the identification of sources of recurrent infections and patient-to-patient transmissions [72]. Despite this enormous potential, the adoption of WGS in clinics is still slow, although new benchtop sequencing platforms have recently been introduced that, being cheaper, could facilitate the spread of bacterial WGS in the clinical setting [72]. 

### 3.9. Next-Generation Sequencing (NGS)

Next-generation sequencing (NGS) is a rapid and cost-effective massively parallel sequencing technology that, by focusing on the exome, is a viable alternative to whole-genome sequencing [73]. In fact, only second-generation NGS should be adopted in clinical microbiology, as it combines short-read technology for output and cost. Second-generation NGS workflows have three key phases: library preparation, template preparation, and sequencing [74]. The main difference between the different NGS platforms concerns the DNA amplification techniques used: sequencing by hybridization and sequencing by synthesis. Both platforms have been successfully applied to outbreak investigations and resistance gene mapping of several ESKAPE pathogens, including *P. aeruginosa* and *A. baumanni* [74].

### 3.10. Microfluidics

The microfluidic “lab-on-a-chip” technique represents a promising technology for the detection of antibiotic-resistant bacteria [75]. Compared to traditional macroscale methods, microfluidics offers many advantages such as: low cost, fast and high-throughput analysis, smaller sample volume, and automation. There are two main categories of microfluidic-based detection methods: genotypic and phenotypic assays [76]. Genotypic on-chip assays (e.g., PCR, LAMP) target genetic markers (e.g., 16S rRNA genes), thus circumventing bacterial growth and allowing rapid bacterial identification. However, this assay is not applicable for the determination of bacterial antibiotic susceptibility profiles. In contrast, phenotypic on-chip assays monitor bacterial growth in the presence of antibiotics, thus offering accurate AST results. Typically, bacterial cells are restricted to a small volume, such as a chamber or droplet, captured by antibodies on membranes or magnetic beads, or encapsulated in chambers containing agarose. The main drawbacks of these microfluidic platforms involve the use of expensive microscopes and the lack of antibodies targeting different bacterial strains. Because of these limitations, improvements are needed to make these systems commercially available [75,76].

### 3.11. Immunodetection of Pathogens

Immunodetection is a simple and specific method used for the identification of microbial pathogens that relies on the use of antibodies which are suitably immobilized on nanoparticles or strips and can bind specifically to a given target. The binding is visualized by labeling the antibody with fluorescent dyes or redox enzymes [28]. In contrast, the direct detection of antibiotic resistance proteins through this technology needs to be implemented.

### 3.12. Detection of Growth-Related Molecules 

A wide range of devices detecting volatile compounds, so-called electronic noses (eNose), have been used in the diagnosis of bacterial infection diseases [77]. The eNose instruments can detect the smellprint patterns of a given bacterial species and discriminate among the complex mixtures of volatile metabolites associated with a specific disease. Several studies have revealed the efficacy of e-nose devices for the clinical diagnoses of many human diseases [78,79].

### 3.13. Biosensor Systems

A biosensor can be defined as a module that aids in detecting changes in physical quantities and thereby converts these changes to signals proportional to the concentration of an analyte in the reaction [80]. Sensors are classified into various categories, depending on the physical substance or analyte to be measured. Several studies showed that using microcalorimetry approaches, it was possible to identify bacteria directly from urine samples and vancomycin-resistant *Staphylococcus aureus* strains in less than 8 h [81,82]. The main disadvantages of these methods are the absence of clinically validated microcalorimetry systems for AST and the need to analyze pure cultures [80].

### 3.14. MALDI-TOF Mass Spectrometry

Matrix-assisted laser desorption/ionization time-of-flight mass spectrometry (MALDI-TOF MS) has become a reference method used for microbial identification in clinical microbiology laboratories [83]. In contrast to traditional methods of bacterial species identification and related susceptibility testing, which are labor- and time-intensive (for example, AST can usually be completed at least 24 h after cultures turn positive), MALDI-TOF is a fast, convenient, and low-cost method for this purpose [83]. The identification of microorganisms using this technique relies primarily on spectral libraries to identify the mass fingerprinting of the peptide generated by each microorganism. Briefly, the bacterial colony is suspended in a matrix solution and placed on the well of a stainless-steel plate. After evaporation of the solvent, the matrix desorbs and ionizes in the presence of a laser beam that transfers energy to the sample molecules, which are accelerated and separated under the action of an electric field according to their mass-to-charge ratio. Microorganisms are identified by comparing the resulting spectrum with a database of spectra of known organisms [83,84]. The most frequently used platforms are the Bruker Biotyper (Bruker Daltonik, Bremen, Germany) and the VitekMS (bioMérieux, Marcy l’Etoile, France). These two platforms have a single set of organisms in their database and both systems have a search-only module that allows the user to add organisms that are not in the database. Both platforms, being equally specific and reproducible, are able to identify the vast majority of organisms commonly isolated in the microbiology laboratory at the species level [85]. The high levels of identification of both techniques are based on the identification of low-molecular-weight spectra ranging from 2 to 20 KDa (i.e., ribosomal proteins). The Bruker Biotyper is better for identifying non-fermenting Gram-negative bacteria and yeasts, while the VitekMS is better suited for the identification of mycobacteria, actinomycetes, and filamentous fungi [85]. Despite the progress, a current challenge of both platforms is the identification of pathogens directly from samples. In addition, another aspect that can significantly develop this technology is related to the ability to identify multiple bacteria in complex polymicrobial samples [86]. In this regard, several processing methods used for the preparation of different clinical samples, such as urine, cerebrospinal fluid, and blood, suitable for MALDI-TOF MS analysis have been proposed and are still in the process of standardization [86,87]. Another critical challenge for MALDI-TOF MS in clinical microbiological diagnosis concerns the detection of antibiotic resistance [88] and the activation of drug efflux pumps [1]. In addition to disk diffusion susceptibility testing, current methods for antimicrobial susceptibility testing include broth dilution methods [1]. The latter are the reference methods for determining the minimum inhibitory concentrations (MICs) of antimicrobial agents (i.e., the lowest concentration at which the agent inhibits the growth of microorganisms). Although accurate, these methods are either labor-intensive or time-consuming, leading to long waiting times to obtain AST results [89]. 

The identification of AMR determinants, mostly based on molecular techniques such as PCR, also has several limitations, the most important of which are: (1) the expression of some resistance genes must be induced; (2) the discovery of new potential AMR determinants is difficult to achieve; (3) identifying the genetic mechanisms of AMR and predicting resistance phenotypes of bacterial pathogens requires a deep knowledge of the field; and (4) these techniques are expensive and time-consuming, which does not make them suitable for wide use in clinical microbiology laboratories [28]. Several approaches based on MALDI-TOF MS have been used for the rapid detection of antimicrobial resistance in bacteria. MALDI-TOF MS has been used to detect the presence of carbapenemases for the rapid detection of antimicrobial resistance [88,90]. The most promising applications include: (1) the determination of β-lactamase activity by visualizing the peak shift of β-lactam ring hydrolysis; (2) the detection of the specific peak for AMR determinants; (3) the detection of specific peaks for proteins co-expressed with AMR determinants; and (4) a comparison of the area under the curve of the MALDI spectra of bacteria incubated with or without antimicrobial drugs [88,90]. 

Although significant strides have been made toward susceptibility determination in antibiotics, several drawbacks need to be resolved such as (1) limited applications directly in clinical samples, (2) expensive equipment, and (3) specific extraction protocols for the detection of biomarkers of antimicrobial resistance [91].

## 4. Conclusions

The rapid diagnosis of serious infections is critical to initiate appropriate therapy as early as possible while simultaneously reducing the use of unnecessary antibiotics when they are not needed and associated morbidity and healthcare costs [1]. The ESKAPE pathogens and other serious health threats, through the rapid acquisition of AMR genes combined with the ability to form biofilms, have the greatest impact on healthcare-associated infections [12]. The main advantages of the molecular approach in the detection of bacterial pathogens and AMR genes are related to (1) obtaining results in reduced time; (2) the direct application of these methods to clinical samples resulting in reduced time-to-response; and (3) cost benefits in terms of reduced time to appropriate therapy and decreased hospitalization and risks associated with co-morbidity and mortality, [30]. Despite recent advances in molecular technologies in the field of microbiological diagnostics, it is not possible to identify a single diagnostic platform that fully meets the clinical need to (1) provide better treatments and care to patients, (2) reduce the use of broad-spectrum antibiotics, and (3) achieve better clinical outcomes.

## Figures and Tables

**Figure 1 pathogens-11-00663-f001:**
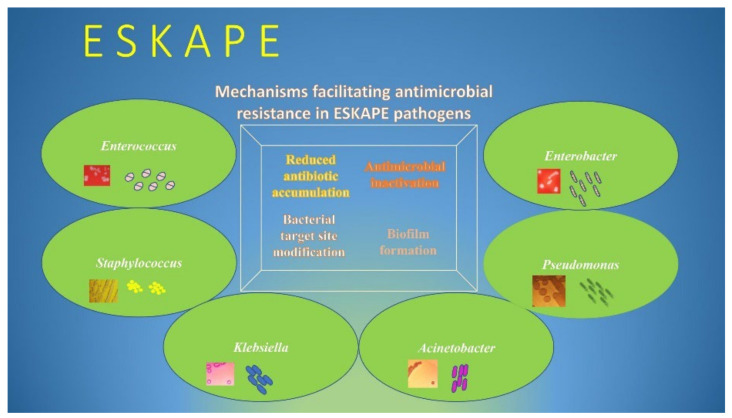
Mechanisms of antimicrobial resistance in ESKAPE pathogens.

**Figure 2 pathogens-11-00663-f002:**
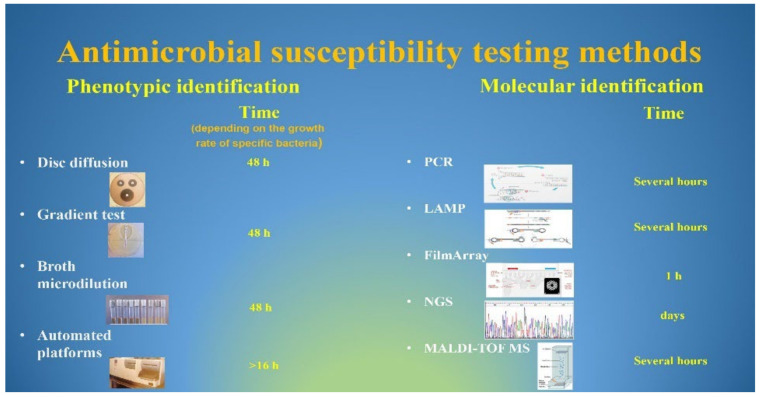
Currently used methods for antimicrobial susceptibility testing.

**Figure 3 pathogens-11-00663-f003:**
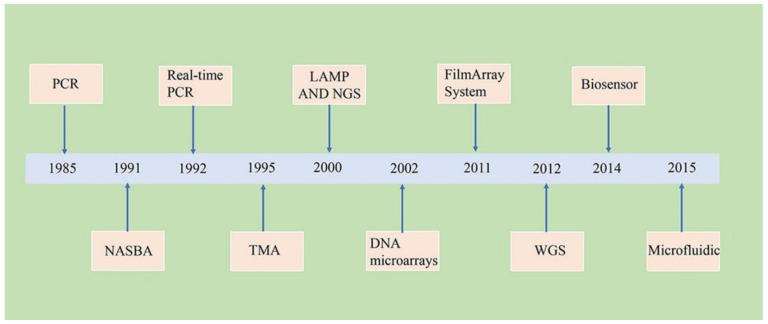
Timeline of major molecular techniques for the diagnosis of bacterial infections.

## Data Availability

Not applicable.

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
