# Peer review of "Recent Advances in the Use of Molecular Methods for the Diagnosis of Bacterial Infections"

_pathogens, 2022, doi:10.3390/pathogens11060663_

Round 1

Reviewer 1 Report

The manuscript is well written but somehow a bit short for a review article. For example, the introduction is entirely about the importance of AMR but no introduction or mention of the main directions of molecular diagnostic, the main subject of the article.  

-It would be very useful to present a table enumerating all the methods, including the main references. 

-It would be also nice to have a timeline figure for the introduction of each method over time. This will provide the reader with an idea of the novelty of each one.  

-I'm missing some emerging technologies for fast detection of AMR. Microfluidics for example, are making a difference as a diagnostic tool. This reference would be a good start: https://pubmed.ncbi.nlm.nih.gov/33672677/ 

-Other important methods are missing: immunodetection of pathogens, whole-genome sequencing, antibiotic degradation products and biosensor systems etc

Author Response

Response to the comments of Reviewer:

We would like to thank the editor and the reviewers since their comments have greatly helped us to improve the manuscript which was carefully revised according to their suggestions. Changes in the revised manuscript are in yellow. (Note: Reviewer comments are in italic, authors' responses are in bold).

Reviewer 1#

The manuscript is well written but somehow a bit short for a review article. For example, the introduction is entirely about the importance of AMR but no introduction or mention of the main directions of molecular diagnostic, the main subject of the article.  

We thank the Reviewer for this important suggestion that has now been incorporated into the revised manuscript

-It would be very useful to present a table enumerating all the methods, including the main references. 

We thank the Reviewer for this important suggestion. A new table has been added in the revised manuscript.

-It would be also nice to have a timeline figure for the introduction of each method over time. This will provide the reader with an idea of the novelty of each one.  

  • Thank you for the suggestion. A timeline figure for the introduction of each method over time has been added in the revised manuscript

  • I'm missing some emerging technologies for fast detection of AMR. Microfluidics for example, are making a difference as a diagnostic tool. This reference would be a good start: https://pubmed.ncbi.nlm.nih.gov/33672677/

We thank the Reviewer for this important suggestion. Now we have added this emerging technology in the revised manuscript as suggested by the Reviewer

  • Other important methods are missing: immunodetection of pathogens, whole-genome sequencing, antibiotic degradation products and biosensor systems etc

We also thank you for these important suggestions that have now been included in the revised manuscript.

Reviewer 2 Report

The authors reviewed the recent advances in the use of molecular methods for the diagnosis of bacterial infections. It is a thorough review that report the most updated methods in diagnosing bacterial infections. Some minor suggestions through out the article can be considered to improve the readablilty:

  1. The paragraphs are too heavy and difficult to read. Please divide into several paragraphs for every section based on the topic discussed in the paragraph. In addition, a lot of long sentences that can break into 2 ~ 3 sentences with period (.) or comma (,) to improve the readability.
  2. Please use comma (,) for numbers. For example, P2 Ln21 54500 hospitalizations can be 54,500 hospitalizations.
  3. Please use scientific description. For example, P2 Ln14 "dangerous" is not scientific and can be replaced with "serious" or other descriptions.
  4. Please use proper English description. For example, nosocomial means hospital-acquired. So nosocomial acquired is not an appropriate description.
  5. Does "CDC" in the article mean "USCDC"? please specify
  6. in P2, why molecular diagnosis can minimize the exposure of lab personnel and this reducing the risk of acquiring infections?
  7.  

Author Response

We would like to thank the editor and the reviewers since their comments have greatly helped us to improve the manuscript which was carefully revised according to their suggestions. Changes in the revised manuscript are in yellow. (Note: Reviewer comments are in italic, authors' responses are in bold).

Reviewer 2#

The authors reviewed the recent advances in the use of molecular methods for the diagnosis of bacterial infections. It is a thorough review that report the most updated methods in diagnosing bacterial infections. Some minor suggestions through out the article can be considered to improve the readablilty:

  1. The paragraphs are too heavy and difficult to read. Please divide into several paragraphs for every section based on the topic discussed in the paragraph. In addition, a lot of long sentences that can break into 2 ~ 3 sentences with period (.) or comma (,) to improve the readability.
  2. We followed Reviewer’s recommendations and we are very grateful for these suggestions. We have now added several paragraphs and improved the readability of the manuscript as suggested by the Reviewer
  3. Please use comma (,) for numbers. For example, P2 Ln21 54500 hospitalizations can be 54,500 hospitalizations.

Done. Thank you for nothing this

  1. Please use scientific description. For example, P2 Ln14 "dangerous" is not scientific and can be replaced with "serious" or other descriptions.
  2. Done. Thank you for nothing this
  3. Please use proper English description. For example, nosocomial means hospital-acquired. So nosocomial acquired is not an appropriate description.

Thank you for the suggestion. The error has been corrected

  1. Does "CDC" in the article mean "USCDC"? please specify

Yes. In the revised manuscript we specify that CDC stands for Centers for Disease Control

  1. in P2, why molecular diagnosis can minimize the exposure of lab personnel and this reducing the risk of acquiring infections?

We agree with the reviewer that the text does not read well and have rephrased it as follows: “Molecular diagnostics can be used to detect microbes directly from clinical specimens, thereby minimizing the occupational exposure of laboratory workers to infectious agents”.

Round 2

Reviewer 1 Report

I endorse the manuscript in its present form. The authors have addressed all my queries.